# Nucleotide-Binding Oligomerization Domain 1 (NOD1) Agonists Prevent SARS-CoV-2 Infection in Human Lung Epithelial Cells through Harnessing the Innate Immune Response

**DOI:** 10.3390/ijms25105318

**Published:** 2024-05-13

**Authors:** Edurne Garcia-Vidal, Ignasi Calba, Eva Riveira-Muñoz, Elisabet García, Bonaventura Clotet, Pere Serra-Mitjà, Cecilia Cabrera, Ester Ballana, Roger Badia

**Affiliations:** 1IrsiCaixa, 08916 Badalona, Barcelona, Spainegarcia@irsicaixa.es (E.G.);; 2Health Research Institute Germans Trias i Pujol (IGTP), Hospital Universitari Germans Trias i Pujol, Universitat Autònoma de Barcelona, 08916 Badalona, Barcelona, Spain; 3University of Vic—Central University of Catalonia (UVic-UCC), 08500 Vic, Barcelona, Spain; 4Centro de Investigación Biomédica en Red de Enfermedades Infecciosas, CIBERINFEC, 28029 Madrid, Spain; 5Pulmonology and Allergy Unit, Hospital de la Santa Creu i Sant Pau, Universitat Autònoma de Barcelona, 08041 Barcelona, Barcelona, Spain; pserram@santpau.cat

**Keywords:** NOD-like receptor, innate immunity, respiratory mucosa, viral respiratory infections, SARS-CoV-2

## Abstract

The lung is prone to infections from respiratory viruses such as Severe Acute Respiratory Syndrome Coronavirus 2 (SARS-CoV-2). A challenge in combating these infections is the difficulty in targeting antiviral activity directly at the lung mucosal tract. Boosting the capability of the respiratory mucosa to trigger a potent immune response at the onset of infection could serve as a potential strategy for managing respiratory infections. This study focused on screening immunomodulators to enhance innate immune response in lung epithelial and immune cell models. Through testing various subfamilies and pathways of pattern recognition receptors (PRRs), the nucleotide-binding and oligomerization domain (NOD)-like receptor (NLR) family was found to selectively activate innate immunity in lung epithelial cells. Activation of NOD1 and dual NOD1/2 by the agonists TriDAP and M-TriDAP, respectively, increased the number of IL-8+ cells by engaging the NF-κB and interferon response pathways. Lung epithelial cells showed a stronger response to NOD1 and dual NOD1/2 agonists compared to control. Interestingly, a less-pronounced response to NOD1 agonists was noted in PBMCs, indicating a tissue-specific effect of NOD1 in lung epithelial cells without inducing widespread systemic activation. The specificity of the NOD agonist pathway was confirmed through gene silencing of NOD1 (siRNA) and selective NOD1 and dual NOD1/2 inhibitors in lung epithelial cells. Ultimately, activation induced by NOD1 and dual NOD1/2 agonists created an antiviral environment that hindered SARS-CoV-2 replication in vitro in lung epithelial cells.

## 1. Introduction

The coronavirus disease 2019 (COVID-19) pandemic, caused by Severe Acute Respiratory Syndrome Coronavirus 2 (SARS-CoV-2), has highlighted how emergent respiratory infections threaten modern societies, causing a deep impact in human health at social and economic levels worldwide. Great advances have been achieved in the management of COVID-19 [1]; specifically, therapeutic anti-COVID-19 vaccines have substantially changed the dynamics of the pandemics in western societies [2,3]. However, the appearance of novel variants with elusive viral escape potential highlights the need for the development of alternative antiviral treatments. Since the beginning of the pandemic, antiviral screening provided candidates the ability to limit the impact of SARS-CoV-2 infection, using ezetimibe, clofazimine or nafamostat, amongst other potential strategies (see [4] for review). However, the currently approved antiviral interventions available are limited to remdesivir [5], molnupiravir [1] and nirmaltrelvir plus ritonavir (Paxlovid^®^) [6,7]. The need for more effective antivirals is also a common thread for other airborne respiratory pathogens, including influenza virus, the respiratory syncytial virus (RSV) or the human metapneumovirus [8]. Common features of therapeutic interventions for respiratory infections include the inaccessibility to the site of infection and the poor induction of a protective immune response in the respiratory mucosa, resulting in poor viral clearance and infection resolution [9,10].

In SARS-CoV-2 infections, the early immune response is critical during the onset of the infection and determines the clinical outcome of COVID-19 disease. Recent studies highlight an inverse correlation between impaired type I interferon (IFN)-induced antiviral responses by plasmacytoid dendritic cells (pDCs) against SARS-CoV-2 and the severity of the disease in severe COVID-19 patients [11,12]. The link between the inability to develop potent type I IFN response and severe pneumonia associated with SARS-CoV-2, but also with influenza and other respiratory viruses, is corroborated by the enrichment of inborn rare variants impairing the activity of innate immune sensing signaling elements [12,13]. Therefore, the improvement of the capacity of the respiratory mucosa to trigger potent immune response at early phases of the infection could be an opportunity for the treatment of respiratory infections. Pharmacological compounds that modulate innate immune sensing and signaling have been proposed as putative cellular targets for the development of novel antiviral therapeutic strategies [13]. Therapeutic approaches targeting evolutionary conserved host factors or host-directed therapies (HDTs) are interesting alternatives to classic direct antivirals. First, HDTs impose a higher genetic barrier to the emergence of resistant strains, and second, HDTs exploit common cellular pathways and processes shared in different families of viruses, having the potential to become broad-spectrum antivirals [14].

In the respiratory mucosa, innate immune cells (e.g., macrophages, dendritic cells, etc.) but also epithelial cells express pattern recognition receptors (PRRs) [15,16] that recognize evolutionary conserved microbial structures, named pathogen-associated molecular patterns or PAMPs. Upon PAMP recognition, intracellular signaling cascades lead to the production of type I IFN, followed by the transcription of interferon-stimulated genes (ISGs). ISG activation results in the release of proinflammatory cytokines and chemokines, which will subsequently promote an antiviral state and pathogen clearance together with the priming of the adaptive immunity [16].

Of note, PRRs are gaining attention in the treatment of respiratory infections, pointing to toll-like receptors (TLRs), RIG-I-like receptors (RLRs) and, to a lesser extent, NOD-like receptors (NLRs) to reduce viral replication (see [17] for review). In viral respiratory infections, intranasal delivery of TLR7 agonist imiquimod has been shown to limit the IAV-associated pathology in mice, and its use in SARS-CoV-2 is still under study [18]. Similarly, STING agonist diABZI as well as RIG-I agonist stem-loop RNA 14 (SLR14), have been shown to limit SARS-CoV-2 infection (including diverse variants of concern) in primary human bronchial epithelial cells and in mice, respectively, by transiently stimulating IFN signaling [19,20,21]. Therefore, accumulated evidence suggests the use of PRR signaling pathways as host-directed therapy to mobilize antiviral defenses.

Here, we explored the capacity of distinct molecules with reported immunomodulatory properties to boost innate immune response in lung epithelial cells with the final objective to identify novel therapeutic targets. In-depth characterization of the most potent molecules resulted in the identification of the NLR pathway as potent and selective inductors of an innate immune response characteristic of antiviral state in lung epithelial cells. Further evaluation demonstrated that innate immune induction specifically by NOD1 agonists also blocked SARS-CoV2 in vitro infection without a broad activation of inflammatory signaling in myeloid cells, suggesting the use of NOD1 agonists as a putative novel antiviral strategy against SARS-CoV-2.

## 2. Results

### 2.1. Assessment of Immunomodulators Able to Activate the Innate Immune System

To evaluate the sensitivity of lung epithelial and myeloid cells to innate immune activation, we screened a group of molecules with reported immunomodulatory potential (described in Figure 1A,B and Appendix A). After a bibliographic review, immunomodulators were selected according to their availability in compound libraries for drug discovery. Immunomodulators are representative of a wide range of PRR sensing pathways, including both agonists and inhibitors of the corresponding targets. Main targeted pathways assessed were those mediated by TLR3, TLR7/8, TLR9, cytosolic DNA sensors (CDSs), RIG-I-like receptors (RLRs), NOD-like receptors (NLRs, e.g., NOD1, NOD1/2 and NOD2) and aryl hydrocarbon receptor (AhR), as summarized in Appendix A, defining their activity as agonist, inhibitor or control for each PRR pathway targeted. To select a subrogate marker for immune induction, cells were treated with the lipopolysaccharide (LPS) positive control, and quantification of IL-8, CXCL10, IL-1β and TNFα was performed by flow cytometry at different time points. Percentage of IL-8+ cells after 6 h of culture was deemed the optimal condition to measure the potency of the selected immunomodulators, as it showed the highest signal under current experimental conditions. Amongst the tested compounds, the distinct NLR agonists targeting NOD1, NOD1/2 and NOD2 pathways showed increased ability to trigger IL-8+ in A549 lung cells and THP-1 myeloid cells compared to other agonists (Figure 1C,D). We observed between 1.5- and 21-fold increase in the percentage of IL-8+ cells depending on the specific NOD agonist and cell type tested. Identified NLR agonists, including selective NOD1, dual NOD1/2 and NOD2 agonists, were selected for further characterization specifically in lung epithelial cells.

### 2.2. Identification of NLRs as Putative Innate Immune Agonists Suitable for the Activation of the Immune Response in the Respiratory Tract

To characterize the ability of NOD agonists to promote an antiviral state in the respiratory mucosa, we screened available selective NOD1, NOD2 and dual NOD1/2 agonists in the lung epithelial A549 cellular model (Table 1).

The ability to elicit a potent innate immune response was determined by the characterization of the cytokine and chemokine profiles in A549 cells. First, we confirmed that none of the NOD agonist compounds impaired cell viability at the tested concentrations in the cellular model of study, measured as the percentage of live cells by flow cytometry (Figure 2, right axis).

Next, in the initial screening, we observed that selective NOD agonists differentially triggered cytokine and chemokine expression in the lung epithelial compartment. In this sense, 50 µM of TriDAP (NOD1) and M-TriDAP (NOD1/2) increased the percentage of IL-8+ cells (3.29- and 3.35-fold, respectively, *p* < 0.05, Figure 2A), compared to the untreated control in lung epithelial A549 cells. Remarkably, NOD1 (TriDAP) and dual NOD1/2 (M-TriDAP) agonists induced up to 2-fold increase in IL-8+ cells compared to the LPS control (Figure 2A). Conversely, in lung epithelial cells, NOD2 agonists did not show significant induction of IL-8 expression, as the percentage of IL-8+ cells remained similar to the untreated condition. No effects were observed for TNFα at any of the tested concentrations in lung epithelial cells (Figure 2B). These results were further confirmed in a dose-response characterization for the best performing agonists. The activity of TriDAP (NOD1) and M-TriDAP, but not MDP (NOD2), increased the percentage of IL-8+ cells in a dose-response manner (Figure 2B, *p* < 0.001 and *p* < 0.05). Altogether, our results indicate that NOD1 and dual NOD1/2 agonists, rather than the NOD2 pathway, selectively promote the expression of cytokines/chemokines in lung epithelial cells.

### 2.3. NOD1 Agonists Selectively Activate the NF-κB and ISRE Signaling Pathways in Human Lung Epithelial Cells

To assess the capacity of the NOD agonists to activate the innate immune response, we used the A549-Dual™ hACE2-TMPRSS2 cell line (hereafter A549-Dual) to study NF-κB and the interferon-sensitive response element (ISRE) activation pathways simultaneously. In lung epithelial A549-Dual cells, treatment with selective NOD1 agonists TriDAP and C12-iE-DAP, as well as dual NOD1/2 agonist M-TriDAP, significantly induced NF-κB pathway compared to the untreated control (*p* < 0.05, Figure 3A). In contrast, the NOD2 agonists, MDP and L18-MPD, elicited a weaker activation of the NF-κB pathway compared to the unstimulated control (*p* < 0.05). Along with the activation of the NF-κB pathway, NOD1 agonists TriDAP and C12-iE-DAP and dual NOD1/2 M-TriDAP also activated the ISRE pathway, as measured by the QUANTI-Luc assay after 24 h of treatment. In A549-Dual cells, NOD2 agonists MDP and L18-MDP were unable to trigger IFN/ISRE activation compared to their NOD1 and NOD1/2 agonists counterpart or the Poly(I:C) positive control (Figure 3B). To further characterize innate immune activation by NOD1 agonists, we assessed interferon-stimulated gene (ISG) expression upon short-term exposure with NOD agonists. First, we confirmed the induction of *IL-8* triggered by TriDAP and M-TriDAP (1.68- and 1.71-fold increase, respectively) compared to the untreated condition after 8 h of treatment (*p* < 0.05, Figure 3C). Expression of the ISGs *CXCL10* and *ISG15* was also upregulated by NOD1 and NOD1/2 agonists. Remarkably, NOD1/2 agonist M-TriDAP potently upregulated *ISG15* expression (148-fold increase compared to ND) at 8 h (Figure 3C). Indeed, M-TriDAP’s effect on *ISG15* expression was more potent than that of LPS and Poly(I:C). Conversely, NOD2 agonist MDP did not modify the gene expression (mRNA) of *IL-8*, *CXCL10* nor *ISG15* at any of the tested conditions. Overall, selective NOD1 and dual NOD1/2, but not NOD2 agonists, activate the expression of NF-κB and ISRE pathways in the lung epithelial cellular model.

### 2.4. Selective Immune Activation by NOD1 Agonist Is Specific for Lung Epithelial Cells

To assess the specificity of NOD1 and/or NOD2 potential interventions at the respiratory tract, we first evaluated the expression of NOD1 and NOD2 receptors in epithelial A549 using myeloid THP-1 cells as control. NOD1 gene expression was observed in both cell types. However, A549 cells did not express NOD2 compared to the THP-1 cells (Appendix A). To confirm the expression of NOD1 in A549, we evaluated its expression at the protein level by immunoblotting. We observed a high protein expression of NOD1 in lung epithelial cells, whilst NOD2 was not detected (Appendix A).

Next, to confirm the specificity of the NOD1 agonist activation pathways, we assessed their activity by transient downregulation of NOD1 gene expression through siRNA in lung epithelial cells. Briefly, transient transfection of A549 cells with siNOD1 inhibited the expression of NOD1 without impairing cell viability (Figure 4A,B). NOD1 downregulation abolished IL-8 activation triggered by NOD1 and dual NOD1/2 agonists TriDAP, M-TriDAP and C12-iE-DAP, respectively, as measured in the percentage of intracellular IL-8+ cells. We observed a 93% decrease for IL-8+ cells in the siNOD1 condition treated with TriDAP or M-TriDAP, together with 86.7% reduction for C12-iE-DAP, comparing to the mock-treated and non-targeting RNA controls (*p* < 0.001 and *p* < 0.01, respectively, Figure 4C).

Moreover, to confirm the specificity of the NOD agonist activation pathways, we also evaluated the activity of NOD1 and NOD2 agonists in the presence of selective NOD inhibitors in lung epithelial cells. First, we observed that incubation with NOD inhibitors abolished the induction of NF-κB and IFN pathways triggered by TriDAP, C12-iE-DAP and M-TriDAP. For the NF-κB pathway, there was a 52% reduction compared to their respective agonist control (Figure 4D, *p* < 0.001). Remarkably, specific NOD1 inhibitor ML130, but not dual NOD1/2 inhibitor (NOD-IN-1), abolished the expression of the NF-κB reporter below the basal activation in cells co-cultured with NOD1 and NOD1/2 agonists (red dot line, Figure 4D). Although a decrease was also observed in the IFN response pathway, ML130 only achieved a significant inhibition in the C12-iE-DAP condition (Figure 4E). Of note, we also observed that dual NOD1/2 inhibitor (NOD-IN-1) reduced the basal activation of the NF-κB and ISRE pathways in ND condition (Figure 4D,E, *p* < 0.01 and *p* < 0.001, respectively), together with a significant inhibition of NF-κB and ISRE pathways induced by LPS and Poly(I:C) controls (*p* < 0.05 and *p* < 0.01). These data suggest that NOD1 and dual NOD1/2 inhibitor could also partially block NLR non-dependent or unspecific activation of the NF-κB and ISRE pathways, as shown for the LPS and Poly(I:C) controls.

Altogether, both experimental approaches resulting in loss of function through gene silencing (siRNA) or pharmacological selective inhibition provide evidence of specific modulation by NOD1 agonist, rather than the NOD2 signaling, to activate the NF-κB and ISRE pathways, promoting cytokine response in lung epithelial cells.

### 2.5. NOD1 Agonists Do Not Elicit Immune Innate Activation in PBMCs

A desired characteristic for an optimal therapeutic intervention is the restriction to the tissue of interest without undesired effects in other compartments. In consequence, we evaluated the effect of NOD agonists in PBMCs as an ex vivo model to evaluate immune system-wide activation. PBMC treatment with compounds targeting NOD1, NOD2 and dual NOD1/2 revealed a different pattern for the intracellular staining of proinflammatory cytokines compared to the lung epithelial model (Figure 5). In PBMCs, all NOD2 agonists stimulated up to 1.67 times the production of interleukin-1 beta (IL-1β), tumor necrosis alpha (TNFα) and interleukin 6 (IL-6) compared to untreated control (*p* < 0.05, Figure 5). Remarkably, dual NOD1/2 agonist PGN-Sandi (20 ng/mL) potency was higher compared to all other agonists and control compounds, as measured as the percentage of IL-1β, TNFα and IL-6 cells by intracellular staining. Conversely to the lung epithelial A549 cells, none of the selective NOD1 agonists induced IL-1β, TNFα and IL-6. We did not observe a dose-dependent response for the NOD2 agonists in PBMCs at the tested concentrations (Figure 5B). However, we confirmed that NOD1 agonists did not impact levels of IL-1β, TNFα nor IL-6 in PBMCs. Taken together, these data may indicate that an NOD1-based intervention would have minimal effect at the systemic level.

### 2.6. NOD1 Agonists Inhibit SARS-CoV-2 Infection in Lung Epithelial Cells

Antiviral activity of NOD1 and NOD2 agonists was evaluated in an SARS-CoV-2 in vitro model of infection using lung epithelial A549-Dual cells and Vero E6. We observed that NOD agonist antiviral activity was dependent on the type of NOD agonist tested. First, NOD1 agonist TriDAP reduced the SARS-CoV-2-GFP+ cells by 49% (*p* < 0.01), whilst dual NOD1/2 ligand M-TriDAP reached up to 57% of infection inhibition (*p* < 0.01), compared to the untreated control. In both cases, the viral inhibition was dose-dependent (Figure 6). Comparing NOD1 to dual NOD1/2 agonists, we observed a 4-fold higher antiviral potency of EC50 TriDAP values to M-TriDAP (Figure 6). Conversely, 50 µM of NOD2 agonist MDP showed limited anti-SARS-CoV-2 activity (25% protection) in relation to the untreated condition. To the extent of our results, dose-dependent antiviral activity of NOD1 (TriDAP) and dual NOD1/2 (M-TriDAP) agonists was significantly more potent compared to the NOD2 agonist (MDP). Finally, we used TLR agonists as controls to activate the innate immunity pathways to block SARS-CoV-2 replication. TLR3 agonist Poly(I:C), a synthetic analog of double-stranded RNA (dsRNA), impaired the viral infection by 45.5% at its highest concentration (*p* < 0.05, Figure 5). Interestingly, the anti-SARS-CoV-2 activity of TriDAP and M-TriDAP was higher than the TLR3 agonist control (Figure 6).

Conversely, NOD agonist did not show any antiviral activity in the IFN-deficient Vero E6 model (Appendix A), further supporting the idea that immune-mediated activation is functionally responsible for the antiviral activity exerted by NOD1 agonists.

## 3. Discussion

The respiratory tract is a particularly vulnerable large surface of the body constantly exposed to respiratory viruses amongst other inhaled pathogens. Viral infections of the respiratory tract often cause a wide range of disease severity, ranging from asymptomatic or mild infection to critical illness that might require intensive care support and even result in high mortality rate due to multiple organ failure. As representative examples of viral infections, SARS-CoV-2/COVID-19 pandemic counts with over 768 million cases and >6.9 million deaths as shown in cumulative data worldwide in 2023 [26]. Similarly, influenza is a century-old disease agent responsible for the seasonal influenza/flu epidemics affecting around 23–30% of children and 10% of adult population with an estimated cost of 290,000–650,000 human lives each year [27]. Unfortunately, there are few effective approved antiviral drugs to limit the impact of respiratory infections, with systematic vaccination campaigns, if available, being the major preventive intervention to reduce the burden caused by respiratory viruses.

Therapeutic approaches targeting host factors to treat viral infections are interesting alternatives. Hence, pathogen-sensing pathways by the innate immune system are interconnected; thus, upon ligand binding to the immune sensors, the signaling cascades often converge in the activation of the same downstream molecular intermediates [13,14]: PRR signaling routes, such as RLRs and NLRs, converge at the MAVS, IKK complex or IKK-related proteins, IRF3, IRF7 amongst other signal transduction elements to subsequently activate transcription factors (NF-κB) and to promote gene transcription. Since viruses from diverse families often exploit common cellular pathways, overlapping therapeutic approaches directed to the immune sensing/signaling level represents an opportunity to boost the immune system upon exposure to pathogens.

In our study, we identified the nucleotide-binding and oligomerization domain (NOD)-like receptor (NLR) family amongst the different PRR subfamilies and pathways tested. NLRs are a family of cytoplasmic PRRs (22 members in humans) that are structurally composed of a carboxy leucine-rich repeat (LRR) domain, which binds PAMP motifs, a central NOD responsible for oligomerization and an N-terminal effector motif responsible for downstream signal transduction. NLRs recognize a broad array of PAMP motifs, such as bacterial cell wall component γ-D-glutamyl-meso-diaminopimelic acid (iE-DAP) by NOD1, muramyl dipeptide (MDP) by NOD2 as well as Poly(I:C) through NLRP3. Upon ligand binding, CARD domain activation recruits the receptor interaction protein 2 (RIP2) to lead to the downstream activation of NF-κB pathway. Alternatively, NLRs have been also described to recognize DAMP signals (e.g., extracellular ATP, uric acid or aluminum hydroxide) associated with endoplasmatic reticulum stress as a result of viral infections such as influenza A virus (IAV) or respiratory syncytial virus (RSV). This non-canonical pathway involves the interaction with the mitochondrial antiviral-signaling protein (MAVS) and the interferon regulatory factor 3 (IRF3)-dependent induction of IFN-α/β for viral clearance [28,29,30,31].

In the lung, NOD1 is expressed in various cell types including lung epithelial cells, endothelial cells, smooth muscle cells and different types of leukocytes [28]. As an NLR, NOD1 has been previously described to actively participate in the host control of bacterial pathogens affecting the respiratory tract, including *Chlamydophila pneumoniae*, *Legionella pneumophila*, *Klebsiella pneumoniae*, *Haemophilus influenzae* and *Pseudomonas aeruginosa* in a mechanism involving, at least, the NF-κB activation pathway [28,29,30]. Moreover, deficient NOD1 expression is associated with impaired *Pseudomonas aeruginosa* bacterial clearance and altered cytokine secretion in NOD1^−/−^ mice [30]. Regarding the antiviral capacity of the NOD1 pathway in the context of viral infections, previous results demonstrated that NOD1 agonist DAP inhibits viral replication of HBV in pretreated C57BL/w mice by decreasing HB antigen and DNA levels due to enhanced T cell activation and immune response [31]. Similarly, NOD1 agonist iE-DAP is known to induce IKKα-dependent phosphorylation and nuclear translocation of IRF3, impairing murine cytomegalovirus (MCMV) replication in mice. Indeed, Fan et al. highlighted how NOD1 exerted a key role of NOD1 for sensing for MCMV but also for human cytomegalovirus [32]. Our findings agree with Yin et al. [33], describing the importance of NOD1, besides MDA5 and LGP2, for an effective innate immune recognition of SARS-CoV-2 and IFN response mediated by interferon response factors 3/5 and NF-κB/p65. Indeed, authors showed how depletion of NOD1, as well as MDA5 and LGP2, drastically reduced the levels of IFN-β upon SARS-CoV-2 infection, rendering lung Calu-3 permissive to viral infection [33]. Although further studies must characterize the specific role of NOD1 in SARS-CoV-2 sensing, Wu et al. [34] postulated a mechanism where NOD1 would promote the IFN production by directly binding viral RNA and modulating the MDA5–MAVS complex formation.

Immune–epithelial cell crosstalk determines responsiveness to viral infection. In our work, NOD1- and NOD2-induced activation is dependent on the model of study: NOD1 receptor is predominantly expressed in lung epithelial cells compared to the NOD2 expression. According to previous studies showing that IFN-I priming of A549 cells mounts an antiviral state upon SARS-CoV-2 infection [35], NOD1-mediated priming might represent an alternative pathway to activate the IFN response in lung epithelial cells as a novel immunoprophylactic approach. Indeed, none of tested NOD agonists lead to a reduction in viral replication in the alternative cellular model Vero E6. Vero E6, a widely used model for viral production, new antiviral discovery and/or drug repositioning studies against SARS-CoV-2, has a high susceptibility to viral infection due to its deficiency in interferon production [36,37]. Taken together, these results suggest that the effect of NOD agonists observed in A549 cells can be attributed to immune activation and the establishment of an antiviral state in lung epithelial cells in response to treatment with the NOD agonists.

Besides NOD1, the modulation of the NOD2 pathway has been also proposed as an antiviral approach: NOD2 agonist MDP was shown to protect against human immunodeficiency virus (HIV) through the NF-κB-dependent expression of proinflammatory cytokines [38]. However, MDP pyrogenic and arthritogenic undesired effects in humans led to the development of MDP derivatives, such as L18-MDP, MDP-LysL18 or murabutide. Murabutide, as a safe synthetic immunomodulator, promotes a non-specific resistance to HIV-1 in macrophages and dendritic cells in vitro challenged with HIV-1 [39,40]. Although further studies are required, the benefits of murabutide administration were suggested in a Phase I/II study with HIV-1 patients presenting weak immune reconstitution and incomplete virus suppression over 2 years on HAART [41]. Similarly, NOD2 pathway antiviral activity has been proposed for a wide variety of viruses, including herpesvirus (HSV), vaccinia virus and, interestingly, in respiratory viruses such as influenza (IAV) [38,42].

A desirable characteristic for a novel antiviral therapy is the specificity at the tissue of interest together with the absence of collateral effects. For an NLR agonist putative approach this means to activate the innate immune response focalizing at the mucosa of the respiratory tract, which is the entry site for respiratory viral infection, without triggering global cell activation and/or inflammatory response. Here, the antiviral activity exerted in lung epithelial cells together with unresponsiveness of PBMCs to NOD1 agonist stimuli suggest a restricted effect of NOD1 agonists at the respiratory mucosal compartment.

Interestingly, NLRs have been shown to limit the timing and amplitude of the immune response. Complications in IAV infection are often associated with the prolonged presence of Th17 cells and neutrophils at the sites of infection, resulting in exacerbated inflammation and tissue damage. In mice challenged with IAV, treatment with MDP (NOD2 agonist) promotes high levels of CXCL12 and CCL5 chemokines to recruit Treg to the lung and concomitant with TGFβ secretion. This anti-inflammatory environment limits the presence of Th17 cells and infiltrated neutrophils to the site of infection, preserving tissue integrity and facilitating infection resolution [43]. Indeed, NOD2 also inhibits TLR7/9 signaling at the IRF7 level as a mechanism of negative regulatory feedback on IFN-I response. The regulatory functions of NLRs are not restricted to NOD2. Other NLRs belonging to the NLRC family, such as NLRX1 or NLRC3, negatively regulate RIG-I-MAVS interplay or STING signaling, respectively [44]. Whether NOD1 might participate in control mechanisms remains to be elucidated. However, since complications in influenza infections and notable in SARS-CoV-2 are related to the excessive production of proinflammatory cytokines and chemokines (“Cytokine storm”), respiratory distress and finally organ failure, it is relevant to explore the dual role of NLR signaling to reduce viremia but also to limit tissue damage.

The limited permeability of the mucosal respiratory tract and the existence of endothelial–epithelial barriers arise challenges for the development of antiviral treatments against airborne pathogens, often requiring the use of high systemic doses of antimicrobial agents through oral or intravenous administration to achieve the necessary concentration in the respiratory lumen [45]. A recent study demonstrated that nasal delivery of remdesivir in African green monkeys resulted in similar levels of pharmacologically active triphosphate in lower respiratory tract tissues, but with a significantly lower dose requirement compared to intravenous administration [46]. This suggests that delivering antivirals via nasal spray could be a viable alternative for reducing SARS-CoV-2 load in the respiratory tract [46]. Therefore, upcoming research needs to overcome the limitations of our findings by confirming the antiviral effectiveness of NOD1 agonists in non-cancerous cell lines, such as overexpressingACE2 BEAS-2b cells [47], and in in vivo studies to establish the appropriate administration routes for early prevention of SARS-CoV-2.

In summary, we identified NLR agonists, amongst the different immunomodulators screened, due to their ability to activate the signaling pathways of the innate immune response through the NF-κB and IFN response pathways. The activation triggered by NOD1 and dual NOD1/2 agonists is limited to the lung epithelial cells, promoting an antiviral environment that prevents SARS-CoV-2 replication. Therefore, NOD1 and NOD1/2 agonist strategies should be explored as host-directed antiviral alternatives to strengthen the innate immune system in the respiratory tract upon early stages of respiratory viral infections.

## 4. Materials and Methods

### 4.1. Cells

Human lung adenocarcinoma cell line A549 and African green monkey kidney epithelial cells Vero E6 cells were obtained from Sigma-Aldrich, Merck (Darmstadt, Germany), and cultured in DMEM medium (Gibco, ThermoFisher, Madrid, Spain) supplemented with 10% heat-inactivated fetal bovine serum (Gibco, ThermoFisher, Madrid, Spain), 100 U/mL penicillin and 100 µg/mL streptomycin. A549-Dual™ hACE2-TMPRSS2 (hereafter A549-Dual) cells expressing Lucia luciferase and SEAP reporters for IRF and NF-κB pathway activation, respectively, and overexpressing human ACE2 and TMPRSS2 were purchased from InvivoGen (San Diego, CA, USA). A549-Dual make up a well-characterized cellular model widely used in in vitro studies in infectious diseases, expressing secreted embryonic alkaline phosphatase (SEAP) and Lucia luciferase reporter genes under the control of NF-κB binding sites and IFN-stimulated response elements, respectively. A549-Dual cells were kept in the same supplemented DMEM medium as the regular A549, in the presence of the selective antibiotics blasticidin (10 µg/mL), hygromycin B gold (100 µg/mL), puromycin (0.5 µg/mL) and zeocin (100 µg/mL) (all purchased from InvivoGen).

Human cell line THP-1 was obtained from AIDS Reagent Program, National Institutes of Health (Germantown, MD, USA) and kept in RPMI 1640 medium (ThermoFisher, Madrid, Spain) supplemented as in A549 cells.

PBMCs were isolated from the peripheral blood of healthy individuals using a Ficoll–Paque density gradient centrifugation method, as previously described [48]. The procurement of buffy coats was conducted through the Catalan Banc de Sang i Teixits. Samples were provided anonymously and without traceable information, with the sole indication of their disease testing status. The PBMCs were cultured in complete RPMI 1640 medium supplemented with 10% heat-inactivated fetal bovine serum, along with 100 U/mL penicillin, 100 µg/mL streptomycin, and IL-2 at a concentration of 16 U/mL before exposure to specific compounds for a period ranging between 48 and 72 h.

### 4.2. Compounds

The following compounds were purchased from Invivogen (Ibian Technologies, Zaragoza, Spain): NOD1 and NOD2 agonist kit (#tlrl-nodkit2), containing C12-iE-DAP, iE-DAP, murmyl-dypeptide (MDP), L18-MDP, M-TriDAP, M-TriLYS, murabutide, PGN-ECndi and TriDAP, were purchased from Invivogen (Ibian Technologies, Zaragoza, Spain); Human TLR3/7/8/9 Agonist Kit (#TLRL-KIT3HW3), containing Poly(I:C) (HMW), Poly(I:C) (LMW), Poly(A:U), Imiquimod, R848, CL075, ssRNA40/LyoVec, ssRNA41/LyoVec, ODN2006, ODN2006control, ODN2216, ODN2216control, ODN2395 and ODN2395control; CU-CPT9a, ODN2088, MRT67307, BX795, BAY 11-7082, Pepinh-MYD, Pepinh-MYD Control, Pepinh-TRIF, Pepinh-TRIF Control, L-Kynurenine, 3p-hpRNA, G3-YSD, G3-YSD Control, H-151, G-140, Indirubin, VACV-70/LyoVec, VACV-70c control, HSV-60, Control for CDS Ligands HSV-60/LyoVec, Poly(dA:dT) LyoVec and Poly(dG:dC) LyoVec. NOD inhibitors NOD-IN-1 and ML130 were purchased from Selleckchem (Munich, Germany). Poly(I:C), lipopolysaccharide (LPS, #L6529) and PMA (#P1585) were obtained from Sigma-Merck, Burlington, MA, USA. All compounds were reconstituted in dimethyl sulfoxide (DMSO) or water according to manufacturer’s instructions and stored at −20 °C until use.

### 4.3. Luminescence and Spectrophotometry Assays

To study the modulation of NF-κB and interferon regulatory factor (IRF) signaling pathways by the compounds of interest, we used the QUANTI-BlueTM and QUANTI-LucTM assays (InvivoGen, Toulouse, France), respectively, according to manufacturer’s instructions. Briefly, 5 × 10^4^ A549-Dual cells/well were seeded in a 96-well plate and incubated for 24 h with the compounds. When using NOD inhibitors, they were added 2 h prior the rest of the compounds. Then, cell culture supernatants were kept for further analysis. For the IRF determination, luminescence was read immediately after adding the luciferase. For NF-κB assay, cell culture supernatants were incubated for 3.5 h, and spectrophotometry was measured with an excitation wavelength of 640 nm. SEAP and luciferase were read in an EnSightTM multimode plate reader (PerkinElmer, Waltham, MA, USA).

### 4.4. Evaluation of Cytotoxicity

Cells were exposed to the compounds of interest for a duration ranging from 24 to 72 h (according to each specific experiment). Subsequently, the cells were subjected to staining for a period of 30 min using the LIVE/DEAD^TM^ Fixable Near-IR Dead Cell Stain Kit (Invitrogen, ThermoFisher Scientific) in PBS, following the guidelines provided by the manufacturer. Alternatively, the identification of viable cells was carried out based on the analysis of forward and side laser light scatter through flow cytometry, as previously described [49].

### 4.5. Flow Cytometry

For intracellular assessment of cytokine production, 1.5 × 10^5^ cells/well were seeded in a polypropylene round-bottom 96-well plate. In the cell lines THP-1 and A549, cells were cultured 6 h in the presence of the compounds at 37 °C and 5% CO_2_, co-incubating with protein transport inhibitors GolgiPlug (5 µL/mL) (#555029, Becton Dickinson, Franklin Lakes, NJ, USA) and GolgiSTOP (0.66 µL/mL) (#554724, Becton Dickinson, USA) for 6 h (THP-1) or the last 4 h (A549 cells). Then, cells were stained with LIVE/DEAD^TM^ as previously described, washed with PBS and incubated at 4 °C with BD Cytofix/Cytoperm (#554722, Becton Dickinson, USA) overnight. In the case of PBMCs, cells were cultured in the presence of compounds overnight, adding protein transport inhibitors two hours after the compounds. Next, PBMCs underwent staining with LIVE/DEAD^TM^, following the established protocol, and were then rinsed and subjected to a 45 min incubation at 4 °C with BD Cytofix/Cytoperm. Then, irrespective of the cell type, all cells underwent two washes with BD Perm/Wash (#554723, Becton Dickinson, USA) solution diluted 1:10 in water. Subsequently, cells were subjected to a 4 °C incubation with the respective antibodies (BioLegend, Palex, Spain): PE anti-human IL-6 (#501107), FITC anti-human IL-1β (#511705) and Brilliant Violet 650 anti-human TNFα (#502938) (at a 1:50 concentration in diluted Perm/Wash) for 1 h. Two additional washes were performed in diluted Perm/Wash, and cells were then suspended in 1% formaldehyde in PBS. The subsequent analysis of the cells was carried out using flow cytometry (BD FACSCelesta™ cell analyzer, BD, Franklin Lakes, NJ, USA) on the same day. Data were then processed utilizing FlowJo™ Software Version 10.6.1. (Becton Dickinson, 2019).

### 4.6. Quantitative RT-Polymerase Chain Reaction (qRT-PCR)

Total RNA was extracted from cell pellets using the NucleoSpin RNA II kit (Ref. 740955, Macherey-Nagel, Düren, Germany) and reverse transcribed using the PrimeScript™ RT-PCR Kit (RR036A, Takara, Kusatsu, Japan) following the manufacturer’s guidelines. Quantification of mRNA levels for all target genes was carried out through a two-step quantitative RT-PCR approach, with subsequent normalization to GAPDH mRNA expression by the DD(ΔΔ)Ct methodology. The primer sets and DNA probes utilized were procured from TaqMan Gene expression assays (ThermoFisher Scientific, Waltham, MA, USA) and included GAPDH (Hs00266705_g1), NOD1 (Hs01036720_m1), NOD2 (Hs01550753_m1), IL-8 (Hs00174103_m1), CXCL10 (Hs00171042_m1) and ISG15 (Hs01921425_s1).

### 4.7. Western Blot

Western blot analysis was performed as previously described [50]. Briefly, treated cells were rinsed, lysed, subjected to SDS-PAGE and then transferred to a polyvinylidene difluoride (PVDF) membrane. The following antibodies were used for immunoblotting: anti-rabbit and anti-mouse horseradish peroxidase-conjugated secondary antibodies (1:5000; Pierce, Dallas, TX, USA); anti-GAPDH (1:2500; ab9485; Abcam, Cambridge, UK); anti-NOD1 (1:1000; #3545; Cell Signaling, Danvers, MA, USA); and anti-NOD2 (1:1000; PA5104317; Invitrogen in milk). Blots were immersed in chemiluminescent substrate (SuperSignal West Pico Plus or Femto, ThermoFisher Scientific), and the signal was visualized using ChemiDoc MP imaging system (BIORAD, Hercules, CA, USA).

### 4.8. Transfection and RNA Interference

Cells were transfected using Lipofectamine™ 3000 Transfection Reagent (L3000001, Invitrogen, Waltham, MA, USA) in 24-well plate following manufacturer instructions for RNAi transfections. Briefly, total of 10 pmol of the corresponding siRNA was transfected in 8 × 10^4^ A549 cells per well. RNA lysates were collected 72 h post-transfection. In parallel, cells were resuspended 67 h post-transfection and re-seeded for the flow cytometry assays. siRNAs used for transfection were ON-TARGETplus Non-targeting siRNA Pool (D-001810-10) and human NOD1 siRNA-SMARTpool (L-004398-00), all from Dharmacon, Waltham, MA, USA.

### 4.9. Virus and Infections

All experimental procedures with virus were conducted at the BSL3 facilities of the Comparative Medicine and Bioimage Centre of Catalonia (CMCiB—IGTP, Campus Can Ruti, Badalona, Spain). Briefly, a ΔORF7a recombinant SARS-CoV-2 virus encoding the GFP gene at the ORF7a locus [51], hereafter SARS-CoV-2-GFP, was kindly provided by Prof. Volker Thiel (Institute of Virology and Immunology, Bern, Switzerland). SARS-CoV-2-GFP was propagated and titrated on Vero E6 cells. For infections, A549-Dual or Vero E6 cells were pre-treated with the compounds of interest for 3 h at 37 °C and 5% CO_2_. Then, cells were challenged with SARS-CoV-2-GFP at a multiplicity of infection (MOI) of 0.01–0.5. Antiviral activity of NOD agonists was measured 24 h (Vero E6) or 48 h (A549-Dual) after infection as measured as the percentage of GFP+ cells by flow cytometry (LSRII, BD). Determinations were performed in triplicates, and data were calculated from three independent experiments.

### 4.10. Statistical Methods

Data were analyzed using PRISM statistical software (Prism 10, version 10.2.3) Unless specified otherwise, all datasets exhibited normal distribution and were presented as mean ± standard deviation. Statistical significance was determined through the use of an unpaired, two-tailed *t*-test.

## Figures and Tables

**Figure 1 ijms-25-05318-f001:**
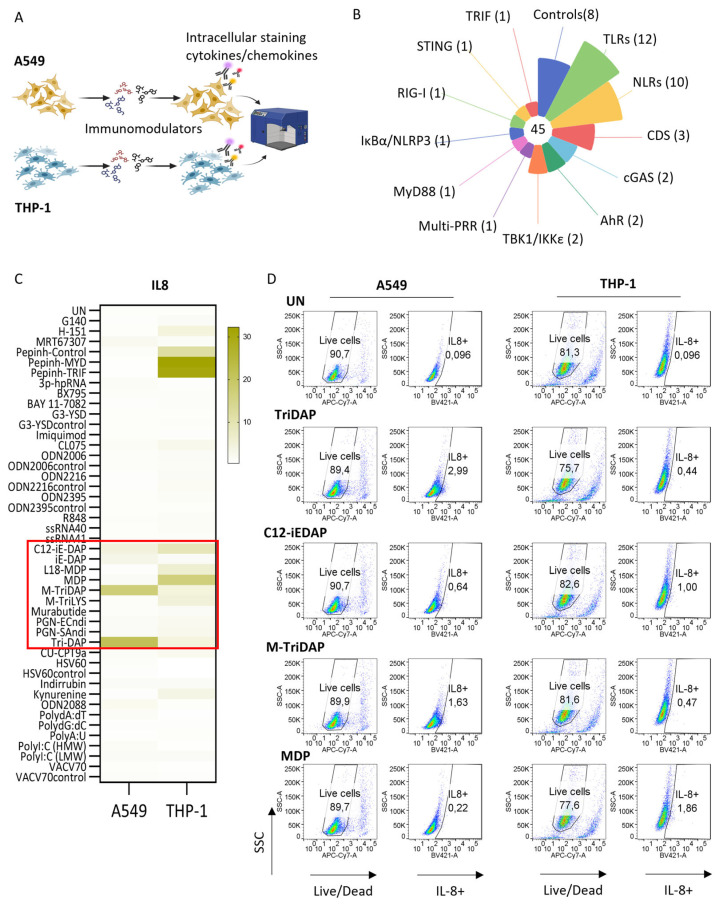
NLR agonists induce innate immune activation in in vitro lung epithelial and myeloid models. (**A**) Workflow to screen for potential immunomodulators of the innate immune system in A549 lung epithelial and THP-1 myeloid cell lines. (**B**) Library classification of tested compounds according to their reported target. (**C**) Heatmap illustrates the immune activation induced by immunomodulators targeting PRR subfamilies in lung epithelial A549 and myeloid THP-1 cells, as determined by the intracellular staining of IL-8 by flow cytometry. (**D**) Representative dot-plots showing IL-8+ intracellular staining of lung epithelial A549 (**left panel**) and myeloid THP-1 (**right panel**) cells upon treatment with NLR agonists, as determined by flow cytometry compared to untreated (UN) cells.

**Figure 2 ijms-25-05318-f002:**
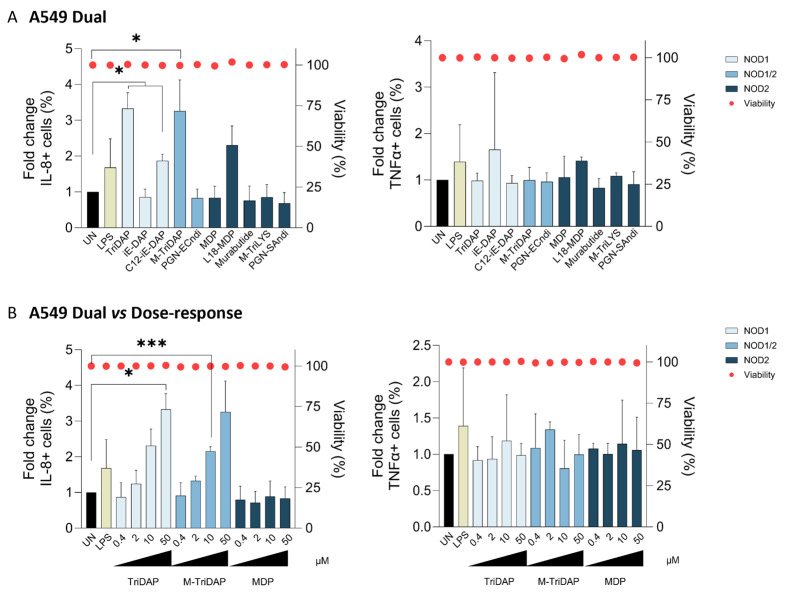
Cytokine response is preferentially triggered by NOD1 and dual NOD1/2 agonists in lung epithelial cells. (**A**) Cytokine response to NLR agonists triggered by NOD1-, NOD1/2- and NOD2-specific agonists in lung epithelial A549-Dual cells. Immune response was determined by the percentage of intracellular IL-8+ (**left**) and TNFα+ (**right**) cell quantification by flow cytometry after 24 h of treatment, using LPS (1 µg/mL, yellow bar) non-treated condition (UN, black bar) as controls. (**B**) Induction of the proinflammatory response upon treatment with increasing concentrations of NOD1, NOD1/2 and NOD2 ligands in lung epithelial A549-Dual cells after 24 h of treatment. The intracellular stainings of IL-8 and TNFα were determined by flow cytometry as subrogate representative markers of the proinflammatory response, using LPS and UN as controls. Mean ± SD of three independent experiments is shown. * *p* < 0.05; *** *p* < 0.001.

**Figure 3 ijms-25-05318-f003:**
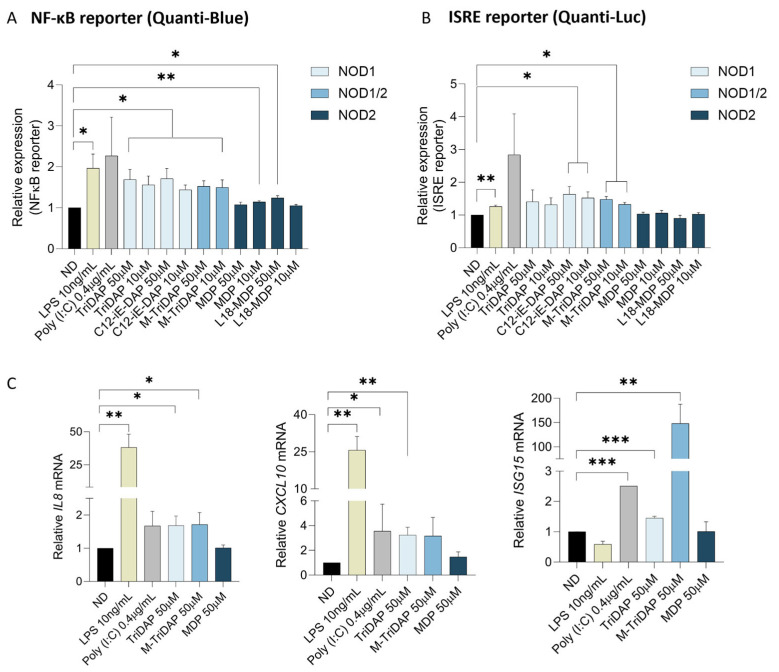
NOD1 and NOD1/2 agonists induce innate immune activation in vitro in lung epithelial through the NF-κB and ISRE pathways. (**A**) Induction of the NF-κB activity triggered by NLR agonists upon recognition by the NOD1, NOD1/2 and NOD2 receptors in lung epithelial A549-Dual cells after 24 h of treatment. LPS (yellow bar) and Poly(I:C) (grey bar) were used as controls for NF-κB activation. (**B**) Assessment of NLR agonist activity on type I IFN response signaling by the quantification of interferon-stimulated response element (ISRE)-dependent gene expression in lung epithelial A549-Dual cells after 24 h of treatment. Values were relativized to the untreated (ND, black bar) condition. (**C**) Relative mRNA expression of IL-8, CXCL10 and ISG15 in A549-Dual treated cells with 50 µM of selected NOD agonists for 8 h measured by qPCR (normalized to GAPDH expression). Mean ± SD of three independent experiments is shown. * *p* < 0.05; ** *p* < 0.01; *** *p* < 0.001.

**Figure 4 ijms-25-05318-f004:**
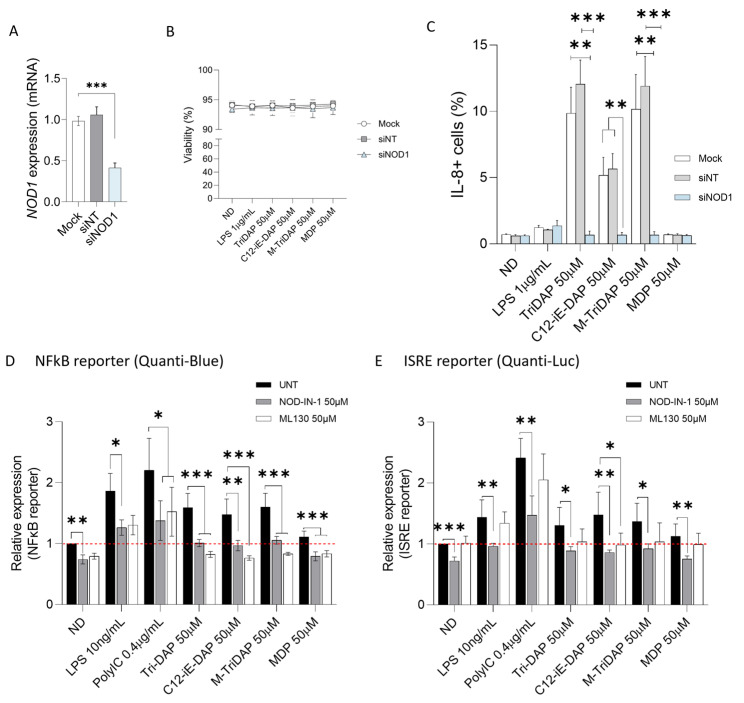
Activity of NOD1 and dual NOD1/2 agonists is specific in lung epithelial cells. (**A**) Gene expression of NOD1 receptor in A549-Dual cells transiently silenced with siRNA targeting NOD1 (siNOD1). Mock and non-specific siRNA (siNT) were used as controls. (**B**) Cell viability of A549-Dual cells treated with siNOD1 and siNT, using mock condition as control. Cell viability was determined by LIVE/DEAD staining and measured by flow cytometry. (**C**) Activity of NOD1 agonists (TriDAP and C12-iE-DAP), dual NOD1/2 (M-TriDAP) and NOD2 (MDP) in A549 cells treated with siNOD1. Intracellular staining of proinflammatory IL-8+ cells was determined by flow cytometry using the mock and siNT conditions, respectively. (**D**) Induction of the NF-κB and ISRE (**E**) activation pathways in A549-Dual cells treated with NOD1, dual NOD1/2 or NOD2 agonists with 50 µM of selective NOD1 inhibitor ML130, 50 µM of NOD1/2 inhibitor NOD-IN-1 or untreated (UNT), respectively. Red dotted line indicates the basal NF-κB (**left**) or ISRE activity (**right**) in A549-Dual cells. Mean ± SD of three independent experiments is shown. * *p* < 0.05; ** *p* < 0.01; *** *p* < 0.001.

**Figure 5 ijms-25-05318-f005:**
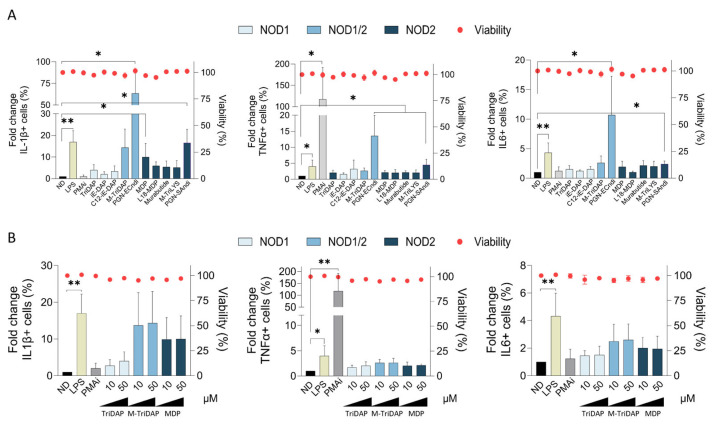
NLR agonist-induced cytokine response is preferentially triggered by NOD2 in PBMCs. (**A**) Assessment of the cytokine response to NLR agonists triggered by specific NOD1, dual NOD1/2 and NOD2 agonists in PBMCs. The percentages of intracellular IL-1β+, TNFα+ and IL-6+ cells were measured as representative markers of the proinflammatory response. Values were relativized to the non-treated condition (ND, black bar). LPS (1 µg/mL, yellow bar) and PMA (50 ng/mL) + ionomycin (1 µM) were used as positive controls. (**B**) Dose-response induction of proinflammatory cytokines IL-1β+, TNFα+ and IL-6+ in PBMCs treated with increasing concentrations of TriDAP (NOD1), M-TriDAP (dual NOD1/2) and MDP (NOD2) agonists. Mean ± SD of three independent experiments is shown. * *p* < 0.05; ** *p* < 0.01.

**Figure 6 ijms-25-05318-f006:**
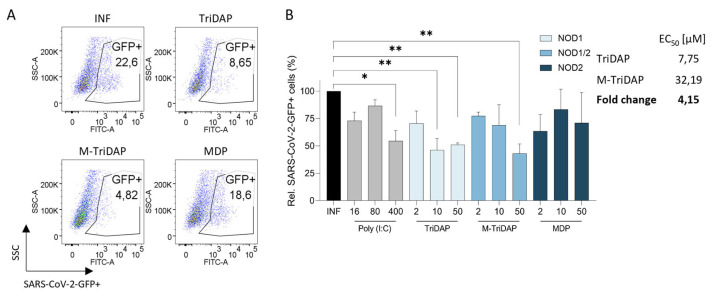
NOD1 and dual NOD1/2 agonists impair SARS-CoV-2 replication in lung epithelial cells. Pretreatment of lung epithelial A549-Dual cells for 3 h with increasing concentrations of NOD1 and dual NOD1/2 agonists preferentially inhibits SARS-CoV-2 replication. (**A**) Representative dot-plots of infected cells treated with NOD agonists as measured by flow cytometry. (**B**) Quantification of viral replication measured as the percentage of SARS-CoV-2-GFP+ cells determined by flow cytometry after 48 h of infection. Values were relativized to the untreated condition (INF, black bar). TLR3 agonist Poly(I:C) (light gray bars) was used as control for the induction of the innate immune response. Mean ± SD of three independent experiments is shown. * *p* < 0.05; ** *p* < 0.01; EC_50_: half maximal effective concentration.

**Table 1 ijms-25-05318-t001:** Summary of NOD-like receptor agonists.

Compound	Full Name	Target	Formula	EC50 ^1^	Cellular Model	Reference
TriDAP	L-alanyl-γ-D-glutamyl-meso-diaminopimelic acid	NOD1	C15H26N4O8	700 ± 100 ng/mL	HEK-Blue hNOD1	[22]
iE-DAP	γ-D-glutamyl-meso-diaminopimelic acid	NOD1	C12H21N3O7	6.3 ± 0.5 ng/mL	HEK-Blue hNOD1	
C12-iE-DAP	Lauroyl-γ-D-glutamyl-meso-diaminopimelic acid	NOD1	C24H43N3O8	170 ± 37.6 nM	HEK-Blue hNOD1	[23]
M-TriDAP	N-acetyl-muramyl-L-Ala-γ-D-Glu-meso-diaminopimelic acid	NOD1/2	C26H43N5O15	30 µg/mL	HCT116	[22]
PGN-ECndi	Insoluble peptidoglycan (*Escherichia coli* K12)	NOD1/2				
Murabutide	N-Acetyl-muramyl-L-Alanyl-D-Glutamin-n-butyl-ester	NOD2	C23H40N4O11	100 ± 20 ng/mL	HEK-Blue hNOD2	[24]
MDP	N-Acetylmuramyl-L-Alanyl-D-Isoglutamine (L-D isoform)	NOD2	C19H32N4O11	146 ± 26 ng/mL	HEK-Blue hNOD2	[24]
L18-MDP	6-O-stearoyl-N-Acetyl-muramyl-L-alanyl-D-isoglutamine	NOD2	C37H66N4O12	0.461 nM	KBM-7	[25]
M-TriLYS	MurNAc-Ala-D-isoGln-Lys	NOD2	C25H44N6O12			
PGN-Sandi	Insoluble peptidoglycan (*Staphylococcus aureus*)	NOD2				

^1^ EC50: Half maximal effective concentration.

## Data Availability

The data that support the findings of this study are available from the corresponding author [R.B. and E.B.] upon reasonable request.

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
