# Peer review of "Nucleotide-Binding Oligomerization Domain 1 (NOD1) Agonists Prevent SARS-CoV-2 Infection in Human Lung Epithelial Cells through Harnessing the Innate Immune Response"

_ijms, 2024, doi:10.3390/ijms25105318_

Round 1
Reviewer 1 Report
Comments and Suggestions for Authors
The manuscript entitled “NOD1 agonists prevent SARS-CoV-2 infection in human lung epithelial cells through harnessing the innate immune response” by Garcia-Vidal et al is focused on the identification of new/alternative immunomodulators to boost the innate immune response in the context of SARS-CoV-2 lung infections.
Here, the authors identified NOD family receptors as new potential targets that, when properly modulated by agonists, can activate the NF-kB and IFN pathways, thus representing an innovative antiviral strategy.
The results reported in the manuscript, validated the approach. Indeed, selective NOD1 agonists were able to activate the immune response in lung epithelial cells. The activity was also specific, since the same effect was not observed in PMBC models, highlighting that the compounds could be deprived of undesired systemic effects. Indeed, while the use of NOD1 agonists seems to be beneficial for activating the immune response during gastrointestinal and respiratory infections, it also seems that NOD1 antagonists should be used in the case of chronic inflammatory disorders, in which NOD are also involved. Thus, the specificity of the reported compounds represents a valuable information.
Most important, the best NOD1 agonist resulted in good anti-SARS-CoV-2 activity, with EC50 of 7.75 µM.
The manuscript is very clear, well written and interesting. For this reason, it is opinion of this reviewer that it can be accepted in the present form.
The few very minor revisions (typo errors), reported below, can be corrected by the authors:
- page 1, line 41, after reference 1 remove the dot and leave the comma;
- page 10, line 354, remove the repetition “of NOD1”;
- in the abstract, please report the meaning of the abbreviation “PRR”.
Author Response
Dear reviewer, here I describe the changes made based on your comments.
- page 1, line 41, after reference 1 remove the dot and leave the comma; Corrected.
- page 10, line 354, remove the repetition “of NOD1”? This repetition was already corrected during the pre-edition changes along the submission process.
- in the abstract, please report the meaning of the abbreviation “PRR”. Corrected, in current abstract, PRR is defined by “Pattern Recognition Receptors before the abbreviation.
Reviewer 2 Report
Comments and Suggestions for Authors
Garcia-Vidal and colleagues address the possibility to counter SARS-CoV-2 infection in lung epithelial cells by activating their intrinsic/innate defense mechanisms using agonists of different pattern-recognition receptors. This strategy may potentially contribute to acute prophylaxis of SARS-CoV-2 infection. Using A549 lung cancer cells as a model and IL-8 production as a readout, they test a broad panel of agonists and find that agonists of NOD1 receptor are most efficient activators of anti-viral defense in epithelial cells. Using luciferase reporter system, they show that NOD1 agonists activate both NF-kB and IRF3 pathways in A549 cells. They also show that effects of NOD1 agonists are relatively specific towards epithelial cells, because mononuclear cells express little NOD1 and poorly respond to its agonists, which will hopefully prevent systemic immune responses if such agonists are administered in vivo. Finally, they show that NOD1 agonists at concentrations 10-50 uM reduce SARS-CoV-2 infection in ACE2-TMPRSS2 transgenic A549 cells by about one-half, being as efficient as poly-I:C at 400 uM.
While these results are interesting and may contribute to coronavirus disease prevention strategies, there are several issues that need to be addressed:
1. A549 are cancerous cells that may not be representative of normal epithelial cells. The anti-viral protective effects of NOD1 agonists should be confirmed in non-cancerous cell lines such as ACE2-transgenic BEAS-2b cells.
2. Activation of type I interferon response by NOD1/NOD2 agonists (in the absence of viral infection) is a controversial issue. In addition to activation of ISRE promoter (which is marginal, honestly speaking), expression of IFNB1 and/or interferon-inducible genes should be measured after treatment with NOD1 agonists. This would strengthen mechanistic explanation of antiviral effects of NOD1 agonists.
3. In the paper by Fan et al (cited by the authors), NOD1 agonists enhanced HCMV-induced IFN-beta response. Did the authors observe the same in their model?
4. How do the authors explain the inhibitory effects of NOD1 and NOD2 inhibitors on LPS- and poly-I:C-induced NF-kB activation (Fig. 4D)?
5. The way of presenting cytokine expression data is somewhat confusing. For example, in lines 114-115, we read “The relative percentage of IL-8+ cells was increased from 1.5 to 21%...”. Does it mean that percentage of IL8+ cells after stimulation was 21% higher than in the control? Obviously, this does not match the values in Fig 2. Or does it mean that percentage of IL-8+ cells was increased 21 times after stimulation as compared to control? This looks more likely, but should be described accordingly in the text.
6. The same concerns Y-axis titles in Fig 2, Fig 4C, Fig 5. They should perhaps be changed to “Fold change of percentage of <cytokine>+ cells. The term “relative percentage” is misleading, anyway.
Comments on the Quality of English LanguageEnglish is of sufficient quality.
Author Response
Dear reviewer,
Here I describe the changes made based on your comments:
1) A549 are cancerous cells that may not be representative of normal epithelial cells. The anti-viral protective effects of NOD1 agonists should be confirmed in non-cancerous cell lines such as ACE2-transgenic BEAS-2b cells.
We appreciate the reviewer’s comment. We agree that our findings would benefit from additional cellular models and we have incorporated the comment and reference in the discussion as a limitation of our study, indicating the need of further studies using non-cancerous ACE2-BEAS-2b cell line, together with in vivo studies to confirm the antiviral activity of NOD1 agonists in the context of SARS-CoV-2 infection (P13; L434-437).
2) Activation of type I interferon response by NOD1/NOD2 agonists (in the absence of viral infection) is a controversial issue. In addition to activation of ISRE promoter (which is marginal, honestly speaking), expression of IFNB1 and/or interferon-inducible genes should be measured after treatment with NOD1 agonists. This would strengthen mechanistic explanation of antiviral effects of NOD1 agonists. In the paper by Fan et al (cited by the authors), NOD1 agonists enhanced HCMV-induced IFN-beta response. Did the authors observe the same in their model?
Thanks for your comment. We have performed additional experiments to evaluate the expression of IFN-inducible genes, together with confirmatory IL8 gene expression in A549-Dual cells with the best-performing NOD agonists (TriDAP, M-TriDAP and MDP at their highest non-toxic concentrations). Although epithelial cells might not respond to stimuli as potently as other cell types (i.e, myeloid cells), we have confirmed the effect of NOD1 and NOD1/2 on IL8. Moreover, we have observed the capacity of NOD1 and NOD1/2 (but not NOD2) to upregulate the mRNA expression of ISG (CXCL10 and ISG15) in lung epithelial cells. We have included these data in the manuscript text (Results section, P6;184-193) and in the Figure 3.
3) How do the authors explain the inhibitory effects of NOD1 and NOD2 inhibitors on LPS- and poly-I:C-induced NF-kB activation (Fig. 4D)?
Thanks for this interesting question. We agree that the result might be counterintuitive. However, although NOD inhibitors are thought to be highly specific (PMID: 21802003), the inhibition of NF-κB pathway in combination with LPS has been previously reported (PMID: 22450316). Moreover, LPS and polyIC are potent activation stimuli signalling through distinct PRRs which complicate the interpretation of these specific results, being its closer investigation out of scope of the present work.
4) The way of presenting cytokine expression data is somewhat confusing. For example, in lines 114-115, we read “The relative percentage of IL-8+ cells was increased from 1.5 to 21%...”. Does it mean that percentage of IL8+ cells after stimulation was 21% higher than in the control? Obviously, this does not match the values in Fig 2. Or does it mean that percentage of IL-8+ cells was increased 21 times after stimulation as compared to control? This looks more likely, but should be described accordingly in the text. The same concerns Y-axis titles in Fig 2, Fig 4C, Fig 5. They should perhaps be changed to “Fold change of percentage of <cytokine>+ cells. The term “relative percentage” is misleading, anyway.
Certainly, the initial cytokine representation was confusing. It was supposed to represent Fold-change as indicated by the reviewer. We have corrected “relative expression” by “fold-change” in the appropriate figures throughout the manuscript and corrected the text accordingly.
Reviewer 3 Report
Comments and Suggestions for Authors
The results are interesting as a new approach as a means that might be used when the lungs are on the verge of infection.
The in vitro experimental technique is excellent and the data is of high value.
I would like to point out a few points that I noticed below.
"L44 To date, few antiviral interventions are available for treatment of SARS-CoV-2 infections"
It would be incomplete without some mention of these agents, which are being researched: Ezetimibe, Clofazimine, and Nafamostat. Some more substances are probably being researched. These drugs have been in use for a long time, so their side effects are well known and they are inexpensive.
in Materials and Methods, the description of screening in Fig. 1 is not clear.
It would be easier to understand if the authors could explain how they did it as a whole, not just the substances and flow cytometry,
It would be better to follow the workflow in Fig. 1A.
"L244 agonists in PBMCs as representative model for a global systemic activation"
The results for PBMCs were clear-cut, but it is questionable, however, whether they warrant a whole-body effect, and the tone of the discussion from L388 should be slightly lowered.
Here the authors only have in vitro experiments, which is the limitation of this study. I respect the authors' efforts and skills, but there are limitations.
Fig.6B
I am sorry to be nit-picky, but doesn't the dose usually get bigger as it moves from left to right?
This is somewhat strange.
These will need to be tested in animals and humans in the future.
I would like to see a discussion on possible means of administering these drugs. Should M-TriDAP be injected intravenously or inhaled, or are there other candidate drugs?
Perhaps these approaches should be initiated very early in the course of the disease. It would be helpful for people if the authors could discuss this as well. Alternatively, would they be effective after the disease has become more severe?
Author Response
Dear reviewer,
Here I describe the changes made based on your comments.
1) "L44 To date, few antiviral interventions are available for treatment of SARS-CoV-2 infections". It would be incomplete without some mention of these agents, which are being researched: Ezetimibe, Clofazimine, and Nafamostat. Some more substances are probably being researched. These drugs have been in use for a long time, so their side effects are well known and they are inexpensive.
We agree your comment. We have stated advances in antiviral screening against SARSCOV2 and added a reference reviewing the subject (P1, L43-46).
2) in Materials and Methods, the description of screening in Fig. 1 is not clear. It would be easier to understand if the authors could explain how they did it as a whole, not just the substances and flow cytometry, It would be better to follow the workflow in Fig. 1A.
The description of the workflow was already included in the Results section (2.1) rather than in the materials and methods. However, we have rephased the sentences of the paragraph to include some details for a better understanding of the Fig 1 – Workflow (P3; L103-119). Thanks to your comment, we have realised that table 1 should be allocated closer to its first mention because in its initial location might end up by adding confusion to the Fig1 comprehension. In our opinion, introduced changes improve the fluency throughout the text for the readers.
3) "L244 agonists in PBMCs as representative model for a global systemic activation". The results for PBMCs were clear-cut, but it is questionable, however, whether they warrant a whole-body effect, and the tone of the discussion from L388 should be slightly lowered. Here the authors only have in vitro experiments, which is the limitation of this study. I respect the authors' efforts and skills, but there are limitations.
Certainly, we appreciate the comment. The conclusions were excessive according to the models used in the article. We have corrected the text in the results section and in the discussion to lower the tone of the conclusions from this part (P9 259-260+L271 and P12; L407).
4) 6B. I am sorry to be nit-picky, but doesn't the dose usually get bigger as it moves from left to right? This is somewhat strange.
We agree the reviewer’s comment. We have corrected the figures of the manuscript to show the concentrations as suggested.
5) These will need to be tested in animals and humans in the future. I would like to see a discussion on possible means of administering these drugs. Should M-TriDAP be injected intravenously or inhaled, or are there other candidate drugs?Perhaps these approaches should be initiated very early in the course of the disease. It would be helpful for people if the authors could discuss this as well. Alternatively, would they be effective after the disease has become more severe?
We agree these comments of the reviewer. We have addressed the suggestions in a single paragraph in the discussion indicating the limitations of our study and the need of further studies with additional models, administration routes and timings to prevent viral replication and subsequent tissue damage (P13; 425-437).
Round 2
Reviewer 2 Report
Comments and Suggestions for Authors
None
Comments on the Quality of English LanguageEnglish is of sufficient quality